# Levels of Empathy in Students and Professors with Patients in a Faculty of Dentistry

**DOI:** 10.3390/bs14090817

**Published:** 2024-09-14

**Authors:** Víctor P. Díaz-Narváez, Joyce Huberman-Casas, Jorge Andrés Nakouzi-Momares, Chris Alarcón-Ureta, Patricio Alberto Jaramillo-Cavieres, Maricarmen Espinoza-Retamal, Blanca Patricia Klahn-Acuña, Leonardo Epuyao-González, Gabriela Leiton Carvajal, Mariela Padilla, Lindsay W. Vilca, Alejandro Reyes-Reyes, Fernando Reyes-Reyes

**Affiliations:** 1Research Department, School of Dentistry, Faculty of Dentistry, Universidad Andres Bello, Santiago 8370133, Chile; joyce.huberman@unab.cl (J.H.-C.); jorge.nakouzi@unab.cl (J.A.N.-M.); chris.alarcon@unab.cl (C.A.-U.); patricio.jaramillo@unab.cl (P.A.J.-C.); maricarmen.espinoza@unab.cl (M.E.-R.); blanca.klahn@unab.cl (B.P.K.-A.); leonardo.epuyao@unab.cl (L.E.-G.); g.leitoncarvajal@uandresbello.edu (G.L.C.); 2Department of Education, School of Dentistry, University of Southern California, Los Angeles, CA 90007, USA; marielap@ostrow.usc.edu; 3Department of Education, South American Center for Education and Research in Public Health, Universidad Norbert Wiener, Lima 15108, Peru; lwquiro@gmail.com; 4Department of Psychology, Faculty of Social Sciences and Communications, Universidad Santo Tomás, Concepción 8320000, Chile; areyesr@santotomas.cl; 5Department of Psychology, Institute of Socio-Emotional Well-Being, Faculty of Psychology, Universidad del Desarrollo, Concepción 4070001, Chile; freyes@udd.cl

**Keywords:** dentistry, education, professor, undergraduate, empathy

## Abstract

Background: Empathy is an attribute that plays an essential role in the dentist–patient therapeutic relationship, clinical care, and treatment adherence, along with providing other benefits. The main objective of this research was to establish the validity, reliability, and invariance of the Jefferson Scale of Empathy and then characterize the empathy levels of students and teachers at a dental school. Materials and Methods: An observational and cross-sectional study analyzed a sample of undergraduate students and professors from the Universidad Andrés Bello Faculty of Dentistry (Chile) (n = 1727 and n = 267, respectively). The Empathy Scale for Health Professionals (HP) and the same scale for students (HPS) were applied. Results: The Jefferson Empathy Scale presents adequate psychometric properties. The empathy measure has adequate reliability and construct validity, confirming a fit of the three-factor empathy model to the data. The measurement is invariant within the university campus, the sex of the student, and between teacher and student. Teachers present greater empathy than students except in the Perspective Adoption dimension. Conclusions: The Jefferson Empathy Scale is reliable, valid, and invariant among Chilean dental students and professors. Students do not differ from their professors in the cognitive component of empathy, but they present a lower score in the affective component and global empathy. It is inferred that students can develop the affective component of empathy in their interactions with their professors, increasing their overall empathy. Understanding and fostering empathy in dental students and professors can significantly improve patient care and treatment adherence and increase patient and dentist satisfaction.

## 1. Introduction

Empathy in healthcare students and professionals towards patients manifests through an intersubjective relationship during dental care [1,2,3,4]. This relationship should also be present between students and professors at any university training future dental practitioners [5,6]. This solidified relationship positively impacts dental patient care, leading to the following: improved communication, responsibility for treatment, reduced fear towards dental students and professors, especially in pediatric patients, and decreased stress and anxiety, among other benefits [7,8,9].

Several instruments measure general empathy and its cognitive and affective components in the general population [10]. However, one of the most widely used instruments to measure empathy in health science students is the Jefferson Scale of Empathy (JES) [8], which has been used and validated preferentially in medical students [11], with less studied versions for health students (HPS) and professionals (HP) [12,13]. These scales apply to all healthcare professionals and measure the empathy levels of professional dentists (including professors) and dental students toward patients.

In Latin America, a hindrance to empathic education is the lack of information on the psychometric properties of measurement instruments. This leads to a biased interpretation of empathy dimensions behavior, which does not reflect the empathic reality of a student or of all students in an educational institution. Accurately estimating empathy is crucial for developing appropriate intervention policies to increase empathy levels through well-founded teaching–learning processes based on observed empathic reality [14]. Studying patient empathy using the JSE instrument (or any of its adaptations to different health science disciplines) implies that observed data must undergo psychometric analyses, including measurement invariance analysis, before proceeding with any other procedure [15,16].

The HP and HPS instruments reveal two empathy components: cognitive and emotional [13,17]. The cognitive component has two dimensions: Perspective Taking (PA) and “Walking in the Patient’s Shoes” (WIPS), while the emotional component consists of one dimension: Compassionate Care [18]. Cutoff points to evaluate empathy and its dimensions in Latin American dental students have been described [15]. These dimensions interact [14,19]; thus, empathy is an integrative attribute of these three dimensions, and individual study of each dimension is only for specifying the development level of each component, appreciating strengths or weaknesses. Consequently, if one dimension “fails” or shows diminished functionality, empathy ceases to be functional or significantly decreases. In extreme cases, empathic functionality failure can lead to psychopathological conditions [20].

The formation of human empathy is complex. This process can be characterized by human species evolution (phylogeny) and surrounding environmental influences from birth to maturity (ontogeny) [21,22]. The development of the neurological system’s structure and function is a phylogenetic process, while complementary development occurs in ontogeny [21,22]. Generally, the limbic system contains the neural basis of emotional functioning, and the prefrontal cortex relates to cognitive functioning [23]. The limbic system’s development can extend until late adolescence (approximately 18 years), influenced by exogenous aspects like family structure, stability, and early childhood socialization of maternal emotions [24]. In contrast, the prefrontal system’s development can extend into early adulthood (25 to 30 years), related to executive functions [25]. Thus, future dental students enter university with a relatively developed emotional structure and limited room for emotional component development. In contrast, some room exists for the learning process of executive functions [5,8,13,16,17,18,21]. Therefore, universities must use these spaces to develop and build empathic training for their students [19].

Studies in various Latin American countries have shown variability in the distribution of empathy levels concerning empathic decline and distribution among groups (between genders, schools, or faculties of the same discipline and between countries). Regarding empathic decline, Hojat et al. [8] proposed a process of declining empathy levels in medical students beginning in their third year. Possible causes include stress [26], ethical erosion [27], loss of physical contact with the patient [28], depression [29], teaching–learning style [30], discontinuity in the teaching–learning process of ethical norms [31], resilience [32], burnout [33], career choice [34], professor’s attitude towards the student [35], and academic and professional climate [36].

Empathic decline has been studied using cross-sectional and longitudinal designs in various disciplines and careers [37,38,39,40,41,42,43,44]. The empathic decline process proposed by Hojat et al. [8] is a particular scientific fact, not a general process [45]. A multicenter cross-sectional study with Latin American dental students showed variability in the behavior of mean distribution between academic courses in dental faculties examined in this study. Such variability does not allow for a unique understanding of empathy decline. More complex studies are needed to understand empathy formation in dental students and health sciences students in general [46]. This phenomenon is also observed in nursing [47] and medical students [14,48]. Variability is also studied between genders, with studies generally establishing that women are more empathetic than men [49]. However, contradictory results exist in samples of Latin American dental students, with equal empathy means and differences in favor of one gender or the other [50].

In Latin America, universities often conduct training activities in multiple cities. A university with different campuses should have the same curriculum, teaching–learning style, and level of additional training for professors. If these premises are met, the outcome of student training in basic, preclinical, and clinical areas should be similar, reflected in a lack of statistical differences. If there are any, they should be few. This phenomenon is called “empathic equalization” [51]. One study related to empathic equalization in Chilean dental students found differences by site (Santiago and Concepción) in the cognitive component of empathy and by gender in both components, cognitive and affective [52].

Some authors coined the term “empathic diagnosis” [53], a set of characteristics describing the empathic situation in an academic unit. This term reflects the reality of empathic behavior in an academic unit, allowing necessary inferences to determine actions to improve or strengthen empathy in students and professors.

We propose to find out what levels of empathy dental students present compared to their professors and if the measure of empathy made by the Jefferson scale is valid, reliable, and invariant when used in a sample of dental students and professors. The aims of the study are (1) to characterize the levels of empathy and its dimensions in students and teachers of the different branches of a dental school and to compare the levels of empathy observed between students and teachers, and (2) to determine the validity, reliability, and invariance of the empathy measurement model, as a prior objective to support the comparisons.

## 2. Material and Methods

### 2.1. Design

Observational and Cross-Sectional [54].

### 2.2. Participants

The examined population is composed of all professors (N = 313) and students (N = 2176) of the Faculty of Dentistry of the Andrés Bello University (Chile). This faculty has three campuses: Santiago, which has 1027 students and 145 professors; Concepción, with 350 students and 68 teachers; and Viña del Mar, with 799 students and 100 teachers. Students complete a twelve-semester program to graduate as dental surgeons at each campus. All students have the same educational model and study program with homogeneous objectives, and the teaching–learning process is based on the same pedagogical strategy. Convenience sampling is used to select a sample of 1727 dentistry students, distributed as follows: Santiago campus = 795 (n1); Concepción campus = 265 (n2); and Viña del Mar campus = 667 (n3). The total number of students sampled by the faculty represents 79.37% of the total population. The distribution by sex was male = 559 (n) (32.37%) and female = 1168 (n) (67.63%). Age was distributed in the sample with a mean = 22.09 and a standard deviation = 2.86. The obtained sample of professors was 267, distributed as follows: Santiago campus = 128 (n1); Concepción campus = 48 (n2); and Viña del Mar campus = 91 (n3). The total number of the sample professors from the faculty represents 85.30% of the total population. The distribution by sex was male = 116 (n) (43.45%) and female = 151 (n) (56.55%). Age was distributed in the sample with a mean = 37.47 and a standard deviation = 7.80.

### 2.3. Instruments

Instruments include the Jefferson Scale of Empathy (JSE) version for health students (JSE-HPS) and version for professionals (JSE-HP) [12,13]. Both instruments are composed of 20 items (10 with positive items and 10 with negative items), and the responses to these questions are structured on a Likert scale with values from 1 (expresses total disagreement) to 7 (expresses total agreement). The HPS and HP scales have three latent dimensions that were described in the introduction: CC (items 1, 7, 8, 11, 12, 14, 18, and 19), PA (items 2, 4, 5, 9, 10, 13, 15, 16, 17, and 20), and WIPS (items 3 and 6). The total possible score is 140 points. This instrument does not diagnose alterations related to empathy, as it is designed to be applied to patients with no mental health problems. The psychometric studies carried out with this instrument show adequate stability and reliability [15].

### 2.4. Procedures

The HP and HPS underwent an adaptation process through a translation and back-translation process (English–Spanish–English). Once this process was completed, the instrument in Spanish was submitted for peer review (five relevant academics from the dentistry surgeon profession, psychologists, and experts in higher education) to establish evidence of cultural and content validity. Subsequently, only the HPS was submitted to a pilot test, with thirty students of both sexes and from different courses, to check the comprehension of the questions. Based on the experience of using the instrument with teachers, it was considered that a pilot study was not necessary.

Data were collected between September and December 2022 at the professors’ regular monthly faculty clinical meetings and in the classroom during regular student theory class sessions scheduled as part of the semester curriculum. In both groups, the instrument was administered in a single session of approximately 20 min in pencil and paper format. The evaluators were professors specially designated by the Department of Studies of the School of Dentistry and by the Directors of each campus because of their experience in applying research instruments. These professors were empowered to use the instrument, answer the participants’ questions, verify that the instrument was entirely responded to before it was handed out, and that the participating students and professors duly signed the informed consent form before answering the empathy scale. The informed consent form was presented to all participants at the beginning of the session, explaining the purpose of the study, the exposure to minimal risk, and the safeguarding of confidentiality. Participants were given sufficient time to read the document and ask questions before signing it. The instrument was administered under license from the Asano-Gonnella Center for Research in Medical Education and Health Care at Thomas Jefferson University. ID: 10905.

All subjects gave informed consent for inclusion before participating in the study. The study was conducted in accordance with the Declaration of Helsinki (2013), and the protocol was approved by the Institutional Ethics Committee of the Universidad Andrés Bello (Approval Certificate: 020/2022, 4 July 2022).

### 2.5. Statistics Analysis

Descriptive analysis. Primary empathy data and its dimensions in students and professors were submitted to normality and homoscedasticity tests using Kolmogorov–Smirnov and Levene tests. Also, statistical means and standard deviation were estimated from the variables of interest.

Fit of the model. The robust Maximum Likelihood (MLR) [55] estimation method was used in the Confirmatory Factorial Analysis (CFA) due to the items showing more than five answer categories [56]. The fit criteria used to evaluate the fit of the model were RMSEA (<0.08), SRMR (<0.08), CFI (>0.95), and TLI (>0.95) [57,58]. The scale’s internal consistency was evaluated through Cronbach’s alpha coefficient [59] and omega coefficient [60]. A value greater than 0.70 is considered to be a value that allows us to verify that the data have internal consistency [61].

The factorial invariance of the scale according to the sex and city of the participants was evaluated through a sequence of hierarchical invariance models: configural invariance, metric invariance, scalar invariance, and strict invariance. The chi-square (Δχ^2^) was used to compare differences in the sequence of models, where no significant values (*p* > 0.05) suggests invariance between groups. In addition, differences in RMSEA (ΔRMSEA) were considered, where differences lower than *p* < 0.015 show the invariance of the model [62].

Comparisons of the empathy levels (and their dimensions) (dependent variables) were made between campus factors and between sexes (independent variables) through bifactorial variance analysis (ANOVA), including the estimation of the interaction between the examined factors [63]. A course or academic year was considered covariable to eliminate its effect on comparing dependent variables. Subsequently, Tukey’s mean multiple comparison test [64] was applied to distinguish the hierarchical relationship between the means. The effect size (ἠ^2^: eta square) of the ANOVA F test, where it is considered 0.01; 0.06; and 0.014 as low, medium, and high, respectively [65], and the power of the test (1 − β) ≥ 0.80. Finally, comparisons between independent samples were made through Student’s *t*-test, considering whether variances of the groups were equal or not equal [63], and the effect size was obtained through Cohen’s d test [65]. Statistical programs used were R 3.3.0+; RStudio 2023.03.01; IBM SPSS 25.0 and G*Power 3.2.9.7. The level of significance used was α < 0.05 and β < 0.20.

## 3. Results

### 3.1. Confirmatory Factorial Analysis (CFA)

Table 1 shows that the model of three related factors presents adequate fit indexes of the total group of university students (RMSEA = 0.038; CFI = 0.934; TLI = 0.924). In addition, it is observed that the model shows adequate fit indexes in each university campus: Santiago (RMSEA = 0.035; CFI = 0.935; TLI = 0.926), Concepción (RMSEA = 0.050; CFI = 0.908; TLI = 0.901), and Viña del Mar (RMSEA = 0.033; CFI = 0.954; TLI = 0.948). In Figure 1, most of the items show moderate factor loads with the factor they belong to. Moreover, factors show significant relationships between them.

Regarding the group of professors, the study found that the model of three related factors presents suitable fit indexes for the total sample (χ^2^ = 267.07; df = 166; *p* < 0.001 CFI = 0.92; TLI = 0.91; RMSEA = 0.051 [IC90% 0.039–0.062]; SRMR = 0.052). Figure 2 shows that these results are similar to those of students.

### 3.2. Measurement Invariance

In Table 1, it can be seen that the factorial structure of the scale has shown evidence of being strictly invariant related to the university campus that students attend. Adequate indicators have been observed in the sequence of proposed invariance models: metric invariance (ΔRMSEA = 0.002), scalar invariance (ΔRMSEA = 0.005), and strict invariance (ΔRMSEA = 0.025). Checking for invariance between two or more interest groups provides evidence that the measures are equivalent in the different groups. This allows students to compare different university sites according to their empathy scores.

In Table 2, it can be seen that the factorial structure of the scale has shown evidence of being strictly invariant according to the sex of participants in the sequence of proposed invariance models: metric invariance (ΔRMSEA = 0.000), scalar invariance (ΔRMSEA = 0.001) and strict (ΔRMSEA = 0.009). It is observed that the scale’s factor structure is strictly invariant when set according to the student’s sex at each university campus (ΔRMSEA < 0.018 in all cases).

Regarding professors, it can be seen in Table 3 that the factorial structure of the scale has shown evidence of being strictly invariant according to the sex of participants in the sequence of proposed invariance models: metric invariance (ΔRMSEA = 0.000), scalar invariance (ΔRMSEA = 0.001), and strict invariance (ΔRMSEA = 0.003).

### 3.3. Reliability Analysis

In this study, it was found that all dimensions of the scale present suitable reliability indexes in the total sample of university students: PA (α = 0.73; ω = 0.72), CC (α = 0.72; ω = 0.73), and WIPS (α = 0.62; ω = 0.62). Similarly, the scale showed suitable reliability indexes at Santiago branches (PA [α = 0.66; ω = 0.67], CC [α = 0.68; ω = 0.69], and WIPS [α = 0.65; ω = 0.65]), Concepción (PA [α = 0.77; ω = 0.76], CC [α = 0.74; ω = 0.75], and WIPS [α = 0.72; ω = 0.72]) and Viña del Mar (PA [α = 0.78; ω = 0.77], CC [α = 0.75; ω = 0.75], and WIPS [α = 0.56; ω = 0.56]). In addition, it shows suitable internal consistency indexes in the men’s sample (PA [α = 0.78; ω = 0.78], CC [α = 0.75; ω = 0.76], and WIPS [α = 0.59; ω = 0.62]), and women’s sample (PA [α = 0.70; ω = 0.70], CC [α = 0.70; ω = 0.70], and WIPS [α = 0.62; ω = 0.62]).

Regarding professors, it was found that all dimensions of the scale present suitable reliability indexes in the total sample: PA (α = 0.83; ω = 0.83), CC (α = 0.72; ω = 0.73), and WIPS (α = 0.73; ω = 0.73). Similarly, this happens in the men’s sample (PA [α = 0.72; ω = 0.71], CC [α = 0.71; ω = 0.71], and WIPS [α = 0.70; ω = 0.71]) and women’s (PA [α = 0.88; ω = 0.88], CC [α = 0.72; ω = 0.73], and WIPS [α = 0.74; ω = 0.74]).

### 3.4. Estimation of Empathy Levels in Students

Table 4 presents the results of estimating the means and standard deviation of the empathy variable and its dimensions by sex in the students.

Table 5 presents the analysis of the variance for comparison of the means of empathy (and its dimensions) of sex, the interaction between campus and sex, and the covariable effect on students and professors.

In the students, the covariable was found to be significant in E, CC, and WIPS, but not in PA. Regarding the campuses, significant differences were found in the CC and WIPS dimensions but not in other dimensions. Significant differences were found in the sex factor in E and all its dimensions. Finally, the relationship between campus and sex factors was significant in the CC dimension. Tukey’s test only showed significance (*p* < 0.05) between the campuses in the PA dimension. It was observed that the mean of Santiago (60.45) are the same as in Concepción (61.3); in turn, the mean of Viña del Mar (62.33) are also the same as in Concepción (61.3), but the mean of Viña del Mar (62.33) is higher than in Santiago (60.45) with statistical differences. Regarding the effect size, small values were observed for its corresponding statistics in the studied factors within each dimension, including those that showed statistical differences. In the case of the power, some factors that had a suitable power were found (>0.80). In contrast, others were low or very low (<0.80). In professors, it was observed that there were no differences between campuses regarding empathy and its dimensions between campuses. Nevertheless, significant differences between sexes in E, PA, and WIPS were observed. Regarding the effect size, small values were observed for its corresponding statistics in the studied factors within each dimension, including those that showed statistical differences. In the case of the power, most of the factors were found to have an unsuitable power (<0.80), though a few were suitable (>0.80). The Tukey test showed no significance (*p* > 0.05) in any of the comparisons made in the present study.

### 3.5. Estimation of Empathy Levels in Professors

Table 6 presents the results of estimating the means and standard deviation of the empathy variable and its dimensions by campus and sex of professors.

### 3.6. Comparison of Empathy Levels between Students and Professors

Table 7 shows the results of the comparisons between professors and students about empathy and its dimensions, considering all faculty as a unit. Statistical differences in E, CC, and WIPS were found. In all cases, the value of empathy is higher in professors. Cohen’s d-test shows that differences tend to be small because all of them are under the range [0.5–0.8] [65].

## 4. Discussion

A review of the literature shows that, in relation to empathy, many studies estimate the levels of empathy and its dimensions, without first testing compliance with the underlying three-dimensional model [14]. This situation may be a methodologic bias. Many authors express the need to evaluate the construct validity, even though the model has been tested in populations similar to those studied [66]. By verifying that the empathy model of three dimensions is valid, we have evidence of construct validity of the empathy measure in the population studied. It has been highlighted that empathy may be influenced by exogenic factors [5] and could have the capacity to change the architecture of one dimension based on the structure of the items that constitute it. Therefore, it is difficult to probe the existence of an agreement between the model and the empirical data on which a determined construct needs to be studied. Verifying the model invariance in several groups guarantees that the number and structure of dimensions are equivalent between groups. This is the only required condition to ensure that the empathy levels and their dimensions can be compared between different groups [67,68]. Research on empathy should then consider this procedure as a demanding methodologic routine.

In this work, the model of the three latent dimensions of empathy was confirmed in students and professors who belong to the same faculty. In addition, the model’s invariance was verified in all the analyzed groups and the data show reliability in general terms and in each studied group.

Overall, studies that try to measure empathy first focus on estimating the descriptive measures of this variable. However, the existence of cut-off points can classify the significance of the observed value as new from a qualitative point of view. In Latin America, such points have been obtained, allowing references concerning the “quality” of the value obtained. A hierarchical cluster analysis with standardized data was used to establish cut-off points in dental students from the Caribbean, Central America, and part of South America. This analysis made it possible to establish three categories of empathy (“high”, “medium”, and “low”) and to construct a norm based on percentiles [69]. In agreement with these cut-off points, E, CC, and PA values observed in this work (Table 4) can be classified as “high” and are around the percentiles 25, 50, and 50, respectively; nevertheless, WIPS value was classified as “medium” around the percentile 25. In general, these results show a higher level of empathy in the emotional dimension, CC, and in one of the cognitive dimensions, PA, compared to the cognitive dimension, WIPS. This dimension is related to the ability to enter a person’s mind and understand their thoughts and feelings, which would be less developed in the sample of students examined. As previously stated, empathy is a three-dimensional system that interacts to originate the empathy attribute. There is a close relationship among dimensions. For instance, some difficulty understanding the patient’s mind may constitute a limitation of efficiently acknowledging one’s thinking [14,19]. This limitation cannot be perceived as the only one; thus, it makes the correct functioning of empathy difficult as a system and may end up affecting the student–patient relationship to some extent, but not critically, because this is not a deficiency but a failure that may be overcome. WIPS as a cognitive dimension can be learned [47]. Therefore, the university can address this issue, including the necessary actions to increase the WIPS levels in its students [70,71,72] during the teaching–learning process. We believe various strategies can be used to improve this aspect in trainee dentists, such as actor-supported simulations of clinical situations with patients, in which trainees have to respond to patients’ emotional and cognitive demands—instances in which real-time feedback can be provided to model expected behavior. Similarly, educational technologies, such as virtual and augmented reality, can create clinical situations, allowing the learner to better understand patients’ thoughts and feelings. The different strategies should maintain continuity throughout the student’s training, allowing for development.

The course factor was used as a covariable for the different comparisons, so it was possible to cancel the variability effect of the empathy levels between courses. For instance, processes such as empathic decline or a sustained increase in empathy and its dimensions throughout the courses [8,27,42,73]. The results of the comparisons between campuses and sexes and the interaction between campus and sex (C*S) allow us to observe that the concept of equalization is fulfilled. This statement is based on whether the statistical significances observed are small or nonexistent (eta squared ≤ 0.027). Additionally, the values for the test power surpass a value of 0.80 in some cases. When it does, it is to effectively infer, in the students’ population, the differences observed (when they exist) between the means of the compared groups are small or non-existent. Nevertheless, it is necessary to highlight that female students had higher values than men in empathy and all its dimensions. The process of accomplishing equalization is complex and depends on many variables that can converge on empathy in the different geographic regions where the campuses from the same university are found. However, the differences in empathy levels between campuses could be explained because the convergence of variables influencing empathy in a geographical region need not be the same for all these regions (considering that between the most distant university campuses, there is a distance of 600 km), and if it were, we should not expect them to behave to the same degree/extent in all areas [74]. The comparison of the equalization studies at universities is difficult because there are no works on such regarding dentistry students in Latin America, with one single exception [52]. The “pure state of equality” (equal levels of empathy across campuses) is difficult to achieve due to the intrinsic variability of the students being assessed. But this difficulty does not prevent empathy as an attribute from being incorporated in a planned manner into the curriculum of a given degree course on all campuses where that degree course is taught. The planning, content and length of the courses, as well as the different university campuses, are the same for all students on the different campuses.

The absence of interactions between campus and sex (C*S) shows that both variables are independent. Thus, the results of comparisons of the empathy levels between sexes are not biased because of the existing interaction.

Studies of empathy in teachers do not provide cut-off points for assessing empathy levels. This is because studies of empathy in dentistry professors and other specialties in Latin America are few [5], and the estimation of this type of cut–off point requires a critical mass of data from different zones of a determined region, and these data must have been obtained with the same methodologic criterion. Therefore, it is difficult to classify the observed results in this work without an existing benchmark. Under these circumstances, the observed values of empathy might be considered high (or higher) in terms of the observed levels in students. This fact, and the absence of differences in empathy levels between campuses, is a positive fact that could build solid empirical ground to continue developing a teaching–learning process for students. Many factors make every teaching–learning process in dentistry formation successful [75,76]. One of them is training and mastery of the subject of professors, along with academic content and skill, didactics, and power of the curriculum [77]. Additionally, there is the applied methodology in any area (basic, preclinic, and clinic) and the teaching of techniques or activities that can be carried out through different strategies; the foundation of the clinical activities; availability at the clinic; professor–student relationship; learning assessment; and the role of the professor in the student’s comprehensive training [78]. The teaching of empathy cannot be structured as the teaching–learning of specific contents related to tangible objects where empiric laws and accurate theories are governed (precise relatively) and must be taught considering the subjective character of the object of study, specifically, the intersubjective relationship between two people [45]. The example a professor sets when teaching, such as regards responsibility, teamwork, ethical commitment, and humane treatment, among others, is a powerful academic tool. This example can be one of the cornerstones of the student’s empathic formation strategy [5,79].

At all campuses, women, in relation to men, presented higher levels of empathy and also higher levels in each of the dimensions of empathy. The observed differences between sexes have been controversial. Some studies state that women are more empathic than men [80], but some studies have not found any differences between the sexes [81]. Latin American authors have found that the distribution of empathy between the sexes varies. After analyzing 18 dental schools, it was observed that in some of them, women were more empathetic than men; in others, there were no differences; and in some, it was observed that men were more empathetic [50]. One consideration for the explanation of this variability can be observed in some studies that have been able to note that men, as well as women, may have different neuronal structures, but the same potential to develop and express empathy, “even though their neuronal pathways and how they express empathy socially can be different” [5,82,83,84]. From these findings, it is possible to agree with some authors that “there is not a consistent pattern to conclude that one genre is more empathic than the other” [5,85]; these ideas are also valid for the observed results in comparisons made between both sexes in students.

Finally, the consistent finding shows that the professors had higher levels of E, CC, and WIPS relative to students, except for in the PA dimension. However, PA and WIPS are dimensions that belong to the cognitive component and may be learning objects for students. Independently, whether the empathy levels found in professors are higher by themselves, the importance of this finding lies in that these differences increase the possibility that professors can massively be an essential part of the structuring and applying of a teaching–learning strategy of empathy.

Although this is not the objective of this work, it is relevant to state that “necessary actions” must be considered, and lead to the construction of a formative strategy for the teaching–learning of this dimension, interacting with other dimensions of empathy. Additionally, it is necessary to highlight that this strategy must be complex given the characteristics of its attributes. This strategy must be applied throughout the formative process, considering that this application is adopting different ways to adapt to the various stages of the formative process in the following so-called areas: “basic, preclinic, and clinic”.

The present study can contribute to developing Empathy Training Programs that seek to improve the dimensions of empathy where students are weaker, better preparing them to interact empathically with patients. Information on empathy levels can be used to develop selection criteria and ongoing training for clinical staff, ensuring they possess high empathic competencies and improving treatment satisfaction and outcomes. The empathy description of an educational community can be used to promote a culture of empathy in the clinical setting.

### Study Limitations

This study could be improved by incorporating other sociodemographic variables in addition to gender, socioeconomic level, age, or other variables associated with the variability of empathy in students and professors. The selected sample represents a private university in Chile. Therefore, the results may not reflect what occurs in state universities, limiting the generalizability of the results. No procedure was implemented to control for social desirability, which could alter the measure’s validity. Finally, being a cross-sectional study, the measurement of empathy at a single point in time may be affected by the particular and transitory current situation of the participants or of the university campus at that moment, which could be remedied with a longitudinal study.

## 5. Conclusions

The Jefferson Empathy Scale presents adequate psychometric properties. The empathy measure has adequate reliability and construct validity, confirming a fit of the three-factor empathy model to the data. The measurement is invariant within the university campus, the sex of the student, and between teacher and student. Students’ empathy levels and the CC and PA dimensions can be considered higher. However, the level for WIPS was classified as low. Therefore, it can be regarded that students may have limitations in establishing an empathic relationship with the patient, possibly due to the lack of WIPS. Empathy and its dimensions can be considered higher in teachers than in students. There are no significant differences in the levels of empathy and its dimensions between university campuses, but there are differences between sexes. Women show greater empathy in the sample of both students and professors. The levels of empathy and its dimensions are generally higher in the professors than in the students of the institution studied, except in the PA dimension. The differences can be considered significant, and it is possible to conclude that this finding is an essential condition for the success of future empathic interventions.

The finding that faculty exhibits higher levels of empathy than students, especially in the affective component, suggests that empathy can be developed through professional experience and education. This implies that dental schools should incorporate structured cognitive and affective empathy training into their curricula. Workshops or role-playing scenarios could be employed, and possibilities for collaboration with disciplines such as psychology and communication could be opened. Understanding and fostering empathy in dental students and faculty could significantly improve patient care, treatment adherence, and patient satisfaction and increase patient confidence. Emphasis on empathy can improve the overall quality of dental education and better prepare students for the complexities of patient interactions.

## Figures and Tables

**Figure 1 behavsci-14-00817-f001:**
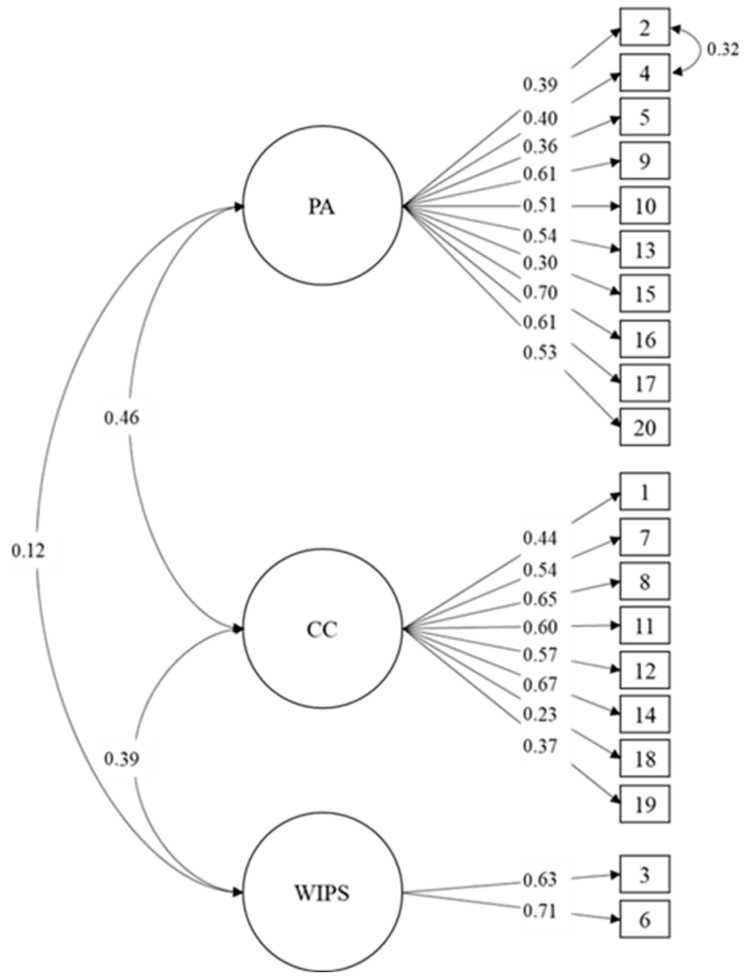
Confirmatory Factor Analysis of the empathy scale in university students.

**Figure 2 behavsci-14-00817-f002:**
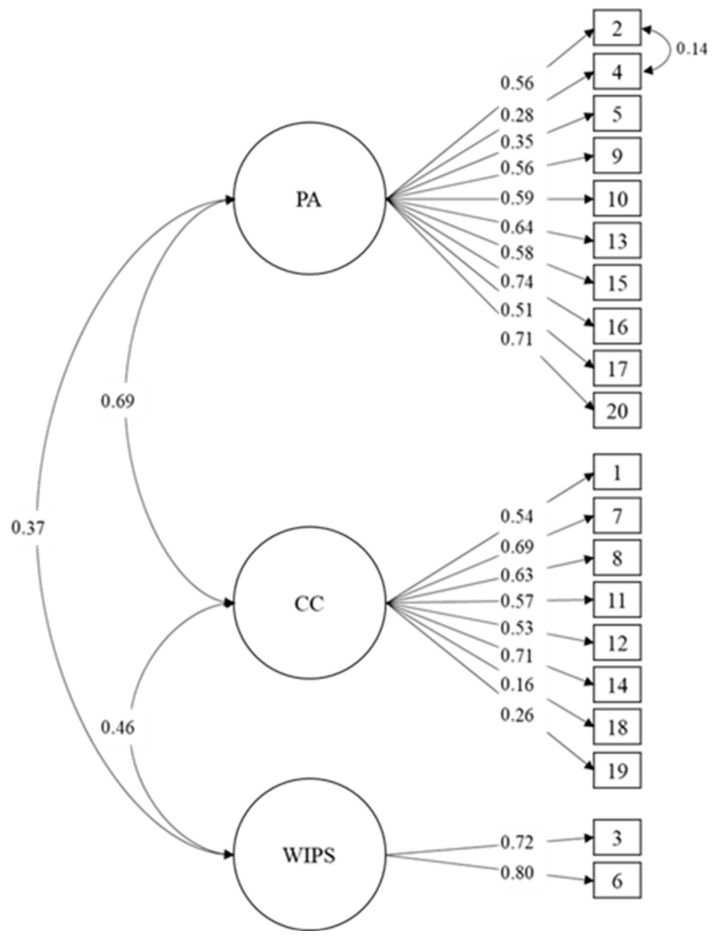
Confirmatory Factor Analysis of the Empathy Scale in Professors.

**Table 1 behavsci-14-00817-t001:** Invariance indexes according to the university campus of the students.

Invariance Models	χ^2^	df	*p*	SRMR	TLI	CFI	RMSEA [CI 90%]	Δχ^2^	Δdf	*p*	ΔRMSEA
Total sample	495.34	166	<0.001	0.035	0.924	0.934	0.038 [0.034–0.041]	–	–	–	–
Santiago	312.35	166	<0.001	0.035	0.926	0.935	0.035 [0.029–0.041]	–	–	–	–
Concepción	264.19	166	<0.001	0.060	0.901	0.908	0.050 [0.039–0.061]	–	–	–	–
Viña del Mar	258.13	166	<0.001	0.040	0.948	0.954	0.033 [0.025–0.040]	–	–	–	–
Configural	831.36	498	<0.001	0.041	0.929	0.938	0.037 [0.032–0.041]	–	–	–	–
Metric	927.41	532	<0.001	0.052	0.920	0.926	0.039 [0.035–0.043]	92.50	34	0.000	0.002
Scalar	1116.99	566	<0.001	0.057	0.896	0.897	0.045 [0.041–0.048]	206.87	34	0.000	0.005
Strict	2019.12	606	<0.001	0.100	0.749	0.733	0.069 [0.066–0.073]	769.88	40	0.000	0.025

Note: χ^2^ = Chi-square; df = degrees of freedom; SRMR: Standardized Root Mean Square Residual; TLI = Tucker–Lewis Index; CFI = Comparative Fit Index; RMSEA = Root Mean Square Error of Approximation; CI = Confidence interval; Δχ^2^ = Differences in Chi square; Δdf = Differences in degrees of freedom; ΔRMSEA = Change in Root Mean Square Error of Approximation.

**Table 2 behavsci-14-00817-t002:** Invariance indexes according to the sex of the students and the sex by university campus.

Invariance Models	χ^2^	df	*p*	SRMR	TLI	CFI	RMSEA [CI 90%]	Δχ^2^	Δdf	*p*	ΔRMSEA
Configural	680.09	332	<0.001	0.039	0.919	0.930	0.039 [0.034–0.043]	–	–	–	–
Metric	713.17	349	<0.001	0.043	0.919	0.926	0.039 [0.034–0.043]	33.20	17	0.010	0.000
Scalar	773.23	366	<0.001	0.044	0.915	0.918	0.040 [0.036–0.043]	65.76	17	0.000	0.001
Strict	1048.43	386	<0.001	0.066	0.869	0.867	0.049 [0.046–0.053]	292.85	20	0.000	0.009
Santiago											
Configural	483.98	332	<0.001	0.043	0.924	0.934	0.036 [0.028–0.042]	–	–	–	–
Metric	513.17	349	<0.001	0.050	0.921	0.927	0.036 [0.029–0.043]	28.29	17	0.041	0.001
Scalar	573.39	366	<0.001	0.052	0.905	0.909	0.040 [0.033–0.046]	66.75	17	0.000	0.003
Strict	873.33	386	<0.001	0.102	0.799	0.796	0.058 [0.053–0.063]	2032.49	20	0.000	0.018
Concepción											
Configural	524.91	332	<0.001	0.077	0.815	0.838	0.068 [0.056–0.780]	–	–	–	–
Metric	556.41	349	<0.001	0.089	0.808	0.824	0.069 [0.058–0.079]	30.56	17	0.022	0.001
Scalar	575.06	366	<0.001	0.091	0.816	0.823	0.067 [0.057–0.078]	18.40	17	0.363	−0.001
Strict	567.93	386	<0.001	0.096	0.838	0.836	0.063 [0.052–0.074]	12.99	20	0.877	−0.004
Viña del mar											
Configural	467.76	332	<0.001	0.049	0.926	0.935	0.038 [0.030–0.046]	–	–	–	–
Metric	478.67	349	<0.001	0.052	0.933	0.938	0.037 [0.028–0.045]	10.83	17	0.864	−0.002
Scalar	497.60	366	<0.001	0.052	0.935	0.938	0.036 [0.028–0.044]	18.01	17	0.388	−0.001
Strict	501.45	386	<0.001	0.056	0.941	0.941	0.034 [0.025–0.042]	17.88	20	0.595	−0.002

Note: χ^2^ = Chi-square; df = degrees of freedom; SRMR: Standardized Root Mean Square Residual; TLI = Tucker–Lewis Index; CFI = Comparative Fit Index; RMSEA = Root Mean Square Error of Ap-proximation; CI = Confidence interval; Δχ^2^ = Differences in Chi square; Δdf = Differences in degrees of freedom; ΔRMSEA = Change in Root Mean Square Error of Approximation.

**Table 3 behavsci-14-00817-t003:** Invariance indexes based on the professor’s sex.

Invariance Models	χ^2^	df	*p*	SRMR	TLI	CFI	RMSEA [CI 90%]	Δχ^2^	Δdf	*p*	ΔRMSEA
Configural	525.03	332	<0.001	0.073	0.842	0.862	0.068 [0.056–0.078]	–	–	–	–
Metric	547.25	349	<0.001	0.083	0.843	0.856	0.067 [0.056–0.078]	23.14	17	0.144	0.000
Scalar	583.47	366	<0.001	0.087	0.837	0.843	0.069 [0.058–0.079]	36.88	17	0.003	0.001
Strict	613.91	386	<0.001	0.100	0.822	0.819	0.072 [0.061–0.082]	31.32	20	0.051	0.003

Note: χ^2^ = Chi-square; df = degrees of freedom; SRMR: Standardized Root Mean Square Residual; TLI = Tucker–Lewis Index; CFI = Comparative Fit Index; RMSEA = Root Mean Square Error of Approximation; Δχ^2^ = Differences in Chi square; Δdf = Differences in degrees of freedom; ΔRMSEA = Change in Root Mean Square Error of Approximation.

**Table 4 behavsci-14-00817-t004:** Results of estimating means and standard deviations of empathy and its dimensions by campus and sex.

Campus	Sex	M (CC)	SD (CC)	M (PA)	SD (PA)	M (WIPS)	SD (WIPS)	M (E)	SD (E)	n
Santiago	Male	42.03	7.375	59.66	7.068	8.12	2.618	109.81	13.073	261
	Female	43.85	6.974	60.83	7.525	8.47	2.768	113.15	12.816	534
	Total	43.26	7.154	60.45	7.395	8.35	2.723	112.06	12.987	795
Concepción	Male	39.22	10.704	60.36	8.364	7.66	3.201	107.24	14.995	105
	Female	44.11	6.831	61.91	6.723	8.49	3.026	114.51	12.325	160
	Total	42.17	8.887	61.30	7.441	8.16	3.118	111.63	13.884	265
Viña del Mar	Male	39.83	9.719	61.02	8.115	7.97	3.121	108.82	15.482	193
	Female	42.97	9.153	62.86	7.742	8.81	3.495	114.64	14.837	474
	Total	42.06	9.420	62.33	7.890	8.57	3.410	112.96	15.245	667
Total	Male	40.74	8.986	60.26	7.704	7.98	2.913	108.98	14.317	559
	Female	43.53	7.921	61.80	7.564	8.61	3.118	113.94	13.618	1168
	Total	42.63	8.379	61.30	7.641	8.41	3.067	112.34	14.036	1727

Note: CC: Compassionate Care; PA: Perspective Adoption; WIPS: “Walking in the Patient’s Shoes”; E: empathy; M: media; SD: standard deviation: n: the sample size.

**Table 5 behavsci-14-00817-t005:** Results of the variance analysis between campuses, between sexes, and estimation of the interaction between campus and sex of students and professors.

Variation Sources	Students				Professors			
	F Test	*p*	η^2^	1 − β	F Test	*p*	η^2^	1 − β
E.								
Course (Cov)	19.211	0.0005	0.011	0.992				
Campus	0.579	0.561	0.001	0.147	0.305	0.738	0.002	0.098
Sex	48.086	0.0005	0.027	1.00	12.592	0.0005	0.046	0.942
Campus by Sex	2.254	0.105	0.003	0.460	0.819	0.442	0.006	0.189
CC								
Course (Cov)	2.514	0.0005	0.013	0.998				
Campus	4.363	0.013	0.005	0.756	0.516	0.597	0.004	0.135
Sex	48.077	0.0005	0.027	1.00	11.050	0.001	0.041	0.912
Campus by Sex	3.388	0.034	0.004	0.639	1.602	0.204	0.012	0.337
PA								
Course (Cov)	2.973	0.085	0.002	0.407				
Campus	8.508	0.0005	0.010	0.967	0.101	0.904	0.001	0.065
Sex	12.596	0.0005	0.007	0.944	4.882	0.028	0.018	0.595
Campus by Sex	0.257	0.774	0.0005	0.091	0.077	0.926	0.001	0.062
WIPS								
Course (Cov)	6.122	0.013	0.004	0.696				
Campus	1.086	0.338	0.001	0.242	0.301	0.740	0.002	0.098
Sex	14.905	0.0005	0.009	0.971	9.980	0.002	0.037	0.882
Campus by Sex	1.152	0.316	0.001	0.254	0.669	0.513	0.005	0.162

Note: CC. Compassionate Care; PA. Perspective Adoption; WIPS: “Walking in the Patient’s Shoes”; E: empathy; Cov: Covariate; η^2^: Partial Eta Squared; 1 − β: Observed Power.

**Table 6 behavsci-14-00817-t006:** Estimating mean and standard deviation of empathy and its dimensions in examined professors.

Campus	Sex	M (CC)	SD (CC)	M (PA)	SD (PA)	M (WIPS)	SD (WIPS)	M (E)	SD (E)	n
Santiago	Male	44.53	7.321	60.32	6.705	10.98	2.899	115.83	12.439	53
	Female	45.64	7.733	62.11	8.674	11.75	2.531	119.49	15.453	75
	Total	45.18	7.556	61.37	7.939	11.43	2.705	117.98	14.345	128
Concepción	Male	42.89	6.781	60.42	6.594	10.37	2.872	113.68	12.763	19
	Female	47.41	7.297	63.14	7.308	12.10	1.970	122.66	11.181	29
	Total	45.62	7.371	62.06	7.090	11.42	2.491	119.10	12.511	48
Viña del Mar	Male	43.91	7.962	60.32	6.107	11.18	2.192	115.41	12.732	44
	Female	48.26	6.819	62.72	9.131	11.96	2.734	122.94	16.464	47
	Total	46.15	7.669	61.56	7.867	11.58	2.504	119.30	15.176	91
Total	Male	44.03	7.448	60.34	6.409	10.96	2.639	115.32	12.515	116
	Female	46.79	7.424	62.50	8.534	11.88	2.490	121.17	15.069	151
	Total	45.59	7.546	61.56	7.744	11.48	2.592	118.63	14.291	267

Note: CC: Compassionate Care; PA: Perspective Adoption; WIPS: “Walking in the Patient’s Shoes”; E: empathy; M: media; SD: standard deviation; n: the sample size.

**Table 7 behavsci-14-00817-t007:** Results of comparing empathy levels and its dimensions between students and teachers considering their faculty as a unit.

Variables	Variances of Groups	F	*p*	*t*	*p*	d
Empathy	Equal variances	1.38	0.244	−6.72	0.0005	0.44
CC	Equal variances	4.087	0.043	−5.39	0.0005	0.37
PA	Equal variances	0.984	0.321	−0.464	0.643	0.03
WIPS	Unequal variances	14.44	0.0005	−17.44	0.0005	0.39

Note: CC. Compassionate Care; PA. Perspective Adoption; WIPS: “Walking in the Patient’s Shoes”; Levene F test; *t*: *t*-Student test, Cohen’s d.

## Data Availability

Data are available on request from the corresponding author.

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
