# Peer review of "Levels of Empathy in Students and Professors with Patients in a Faculty of Dentistry"

_behavsci, 2024, doi:10.3390/bs14090817_

Round 1

Reviewer 1 Report (New Reviewer)

Comments and Suggestions for Authors

General

The study aimed to assess empathy levels in students and professors and also to determine invariance of the empathy measurement model. The paper focuses more on the measurement model, with less emphasis on the results of the study themselves. I think it may have been more appropriate to subdivide the study into two papers so that both objectives could be adequately addressed. The introduction is very lengthy and lacks focus. For example, it contains detailed information about the model which would have been more appropriate in the methodology. It does not provide sufficient background context about the importance of empathy, its emphasis in dentistry, the rationale for its measurement. It also needs to synopsise what has been found in previous studies. Overall the methodology is sufficient but some further detail is required in terms of data collection. The format of presentation of the results requires improvement. It is repetitive in terms of the way the student and professor results are presented (using almost identical paragraphs), with a significant amount of statistical information given in the text and lot of detailed tables. The format of presentation needs to be revised to help improve the flow of the paper and facilitate interpretation of results. The discussion section could be structured better, with use of subheadings. The conclusions require implications plus concluding comments in terms of the validation process. I have given comments in detail facilitate the authors in developing a revised submission.

 Abstract

The abstract does not refer to the measurement tool assessment which is a core component of the study. The abstract should reflect the core content of the paper.

Page 1 line 23-28

The results are poorly described and the conclusions are weak, with no implications provided in terms of the key issues identified in the study.

1. Introduction

Page 1 line 32-39

The introduction should provide a definition of what empathy is and why it is so important both for patient care and also for professors in terms of their interactions with students. Why is empathy important to patient care and is it something that can be further developed through training? These are the types of issues that need to be addressed. This section needs to be reworked to include these issues.

Page 1 line 40-41

The Jefferson scale is introduced as if it is the only scale available. This is not the case. A quick look at the literature has established a number of tools. As such the text should be reworded to reflect this, and to explain why this tool was chosen. If the Jefferson scale has already been validated, the justification for the focus here on validating the tool should be given. Is it that the tool had never been applied in Spanish, or is it that the three components of the model have not been validated? This need to be clarified.

Lima, F. F. D., & Osório, F. D. L. (2021). Empathy: assessment instruments and psychometric quality–a systematic literature review with a meta-analysis of the past ten years. Frontiers in psychology12, 781346.

Moya-Salazar, J., Goicochea-Palomino, E. A., Porras-Guillermo, J., Cañari, B., Jaime-Quispe, A., Zuñiga, N., ... & Contreras-Pulache, H. (2023). Assessing empathy in healthcare services: a systematic review of South American healthcare workers’ and patients’ perceptions. Frontiers in Psychiatry14, 1249620.

Page 1 line 40-page2 line 58

The detailed description of the tool would be better placed in the methodology. The description of the interaction effect needs to be reworded to make it easier to understand.

Page 2 line 59-84

This is far too detailed and needs to be summarised. The role of the university in terms of further developing and building empathy needs to be emphasised.

Page 2 line 85- page 3 line 101

This paragraph is poorly worded and the issue the author is trying to raise is difficult to determine. This paragraph should be reworded.

Page 3 line 3 102

Rephrase ‘on the other hand’

Page 3 line 3 102-135

This paragraph is too long and needs to be summarised. It does introduce issues of interest such as emphatic decline and gender differences but nevertheless needs to be condensed. For example, these variables are not included in the analysis.

Page 3 line 136-138

Supporting references of studies of professors and of students should be provided here.

Page 3 line 139-page 4 line 152

I don’t think this is relevant here and should be removed. I think a sentence about empathetic equalisation can be given in the methods in describing the sample choice and the assumption that it is assumed the teaching style and curriculum is uniform.

Page 4 line 153-157

Testing the invariance  of the empathy measurement model is not included in the abstract, yet it appears to be the main focus of the paper. This needs to be rectified as the abstract is misleading.

2. Materials and Methods

The materials and methods section contains a lot of sub headings with ltd text. I appreciate that this may be the journal style, but if it is not, it would be more user friendly to merge th information under broader headings.

Page 4 line 162-163

Clarification of the population needs to begiven. This this comprise all staff (including administration staff, research staff etc) or just professors? Does it include all students registered full time and part time and all college years? all students and all professors?

Page 4 line 169-170

Could a line about empathetic equalisation be included here. In addition provide detail about the pedagogical strategy in terms of how empathy is developed.

Page 4 line 171-173

The number of variables included in the analysis is small. Surely, other sociodemographic variables to investigate factors affecting empathy such as those raised in the introduction (e.g. empathetic decline) should have been investigated. For example college year? If the list of variables cannot be expanded then this should be raised as a limitation.

2.4- Page 4 line 174-176

More information needs to be provided in terms of how the data was collected. This could be merged with 2.7 and rephrased “data collection”

2.5- Page 4 line 177-180

This needs to be reworded- e.g. ‘applied?’ and this could be merged with 2.6

2.7- Page 4 line 184-191

This needs to be reworded- e.g. ‘empowered to apply the instrument’. In addition further information in terms of administration should be given. Was the form completed during the class, at the end of the class, was it submitted at the end of the class or was it posted or handed in at a later date. How were the responses from the participating professors obtained?

Page 5 line 195

Rephrase ‘criteria of judges’

Page 5 line 197

The rationale for not piloting the professor form should be provided.

Page 5 line 199

Rephrase ‘made up’

3. Results

Page 6 line 242-257

This is an example of the repetitive reporting I referred to earlier. This form of reporting throughout the results should be reworded.

Page 6 line 263

Rephrase ‘on the other hand’- I don’t think this wording is appropriate in an academic paper.

Page 7 line 270

Rephrase sentence – e.g. ‘ones’

Page 8 line 273- to page 10 line 318

The reporting style needs to be improved as it is somewhat repetitive. The findings should be presented in a way to facilitate interpretation of results by synthesising the key findings. For example, the findings showed in measurement invariance across all variables investigate both for students and professors (e.g. campus and sex and also sex in each university campus). The need to present so much statistical information should be reviewed.

3.4- page 10 line 319-339

Needs some rephrasing- e.g. ‘in the group of men’ and ‘in the men sample’

Page 12 ine 350

Reword ‘but the rest’- ?

4. Discussion

The discussion could be structured better involving the use of subheadings, grouping into issues concerning the validation of the model and its conponents, and then the empathy results themselves.

Page 13 line 392-393

Supporting references required here.

Page 14 line 397

Rephrase ‘a sure guarantee’

Page 14 line 402

Rephrase ‘on the other hand’

Page 14 line 415

Rephrase ‘which allow to have references’

Page 14 line 424

Rephrase ‘downward trend.’ Also how do WIPs compare to other studies? This is important as it may reflect an issue within dentistry generally, or alternatively may be an issue specific to the university that was the focus of the study.

Page 14 line 435

Although supporting references are given, examples of the teaching actions that may be effective in improving WIPS should be given in the text, from dentistry and from other disciplines.

Page 14 line 437-440

This needs to be reviewed as it is somewhat disrupts the flow of the discussion.

Page 15 line 454-456

Rephrase ‘should not wait that’ and ‘are no works as such’. Can we not learn anything from the one study that you found?

Page 15 line 466-467

This sentence needs rephrasing.

Page 15 line 472-473

This sentence needs rephrasing.

Page 15 line 478

Use of capital T ‘The applied’

Page 16 line 516

‘as such’

4.1 Page 16 line 520-522

Surely more limitations can be provided- e.g. the variables included in the analysis, no measurement of empathetic decline, the way the survey tool was administered etc.

5. Conclusions

The numbering format of the paragraphs is inconsistent with the style employed in the rest of the paper. The conclusions should give implications in terms of issues such as policy and curriculum development. No conclusions about the validation process are given, yet this comprised a large component of the analysis.

Comments on the Quality of English Language

Although the standard of English in the paper is very good, the phrasing of some of the sentences is a little unusual, and the points the authors are trying to make are difficult to interpret. I do not think that it requires the services of a language translator or specialist editor, but I do think the whole paper needs to be reviewed in detail in terms of the phrasing. 

Author Response

Reviewer 2 Report (New Reviewer)

Comments and Suggestions for Authors

The article is well-written, coherent, and provides new data. 

The introduction is quite lengthy and should be condensed.    It covers multiple subjects that, while interesting, are beyond the primary focus of the study.

Additionally, the research question is not explicitly stated, so the authors should make it explicit.    This is crucial because in the materials and methods section, the authors first describe a strategy to validate the translation of the tool into Spanish, creating ambiguity regarding whether the study's objective is to validate the tool or to gather data about the level of empathy among the surveyed students and teachers.

The statistical analysis is robust and well-designed.

However, there is similar confusion in the results section, where the first part describes the validation results of the instrument and the second part describes the level of empathy.    It's unclear whether the study is a validation of a tool or not.    If it is, the authors will need to change the article's title

Likewise, the discussion is extensive and, in some parts, very argumentative, moving away from a directed and specific discussion focused on the results obtained. 

On the other hand, it is recommended to punctually and emphatically explain how the results can be useful in clinical dental care and whether they can be applied immediately.   

Round 2

Reviewer 1 Report (New Reviewer)

Comments and Suggestions for Authors

Author Response

Comments to authors: levels of Empathy in Students and Professors with Patients in a Faculty of Dentistry R1

Several corrections have been made, having shortened some paragraphs of the introduction and increased the text in the discussion and conclusion; the modifications essentially made are commented on and highlighted in red letters in the corrected text)

Abstract

 Page 1, line 18

Should refer to the Jefferson empathy scale.

Response: Page 1, lines 17-19 is declared and highlighted:

The main objective of this research was to establish the validity, reliability, and invariance of the Jefferson Scale of Empathy and then characterize the empathy levels of students and teachers at a dental school.

Page 1, line 27

The conclusions remain weak, with limited implications provided in terms of the critical issues identified in the study.

Response: Page 1, lines 27-33 is declared and highlighted:

The Jefferson Empathy Scale is reliable, valid, and invariant among Chilean dental students and professors. Students do not differ from their professors in the cognitive component of empathy, but they present a lower score in the affective component and global empathy. It is inferred that students can develop the affective component of empathy in their interactions with their professors, increasing their overall empathy. Understanding and fostering empathy in dental students and professors can significantly improve patient care and treatment adherence and increase patient and dentist satisfaction

1.  Introduction

 Page 1, lines 33-41

This needs to focus on empathy towards patients since this is what the tool measures. There is also a need to state whether the development of empathy is part of the training curriculum.

Page 2, line 42-46

This is better but the rationale for testing the reliability and validity of the tool needs to be given. If it is the most widely used instrument, is there a need to test its validity? There is a need to integrate with p2 lines 79-89.

Response: Page 2, lines 46-50, the change was made.

Page 2, lines 79-89

Response: Integrate with p2 lines 53-62.

The paragraph was changed when the introduction was summarized, and this expression is no longer present.

2.  Materials and Methods

Page 3, lines 183-192

Were forms collected at the end of classes, or were they posted?

Response: Page 3, lines 179-191 highlights:

The procedure was detailed to make it more transparent, but how it is laid out seems sufficient. The following is highlighted in red

4.  Discussion

Page 12, line 377

This still needs rephrasing

Response: Page 12, lines 374-378. The paragraph is rewritten, focusing the change on the sample of students tested

Page 12, lines 386-387

My original comment has not been addressed (Page 14, line 424 of the original manuscript)

Response: Page 12 y 13, line 387-394

Broadens the scope and provides examples in the context of dental training.

Page 13, line 403

This sentence still needs to be rephrased.

Response: Page 13, line 408-412

The text is slightly modified to express the idea better.

Page 15, lines 466-467 (original manuscript)

Please check. You have not given the new page and line reference, and it appears unchanged to me.

Response: Page 13, line 424. the sentence is amended as follows

Page 14, lines 468-473

The limitations are extended with information that is not a limitation but rather.

  ot a limitation. There has been only one line with an additional limitation. The limitations need to be further expanded.

Response: Page 14, lines 475-491. The limitations are expanded, and the decontextualized paragraph is moved to the body of the discussion.

5.  Conclusions

Implications in terms of issues such as policy and curriculum development should be given.

Response: Page 15, lines 508-517. Implications arise, which are developed in the conclusions section.

Reviewer 2 Report (New Reviewer)

Comments and Suggestions for Authors

The authors followed the suggestions and recommendations, but the introduction is still long and contains data that, although interesting, may not be essential for framing the study. It is strongly recommended to reduce it. 

Author Response

The introduction is summarized and responses are given to the various comments raised.

Round 3

Reviewer 2 Report (New Reviewer)

Comments and Suggestions for Authors

The authors have addressed all the suggestions and comments requested; therefore, my recommendation is approved for publication. 

This manuscript is a resubmission of an earlier submission. The following is a list of the peer review reports and author responses from that submission.

Round 1

Reviewer 1 Report

Comments and Suggestions for Authors

Dear. Author,

This observational, cross-sectional study assessed the level of empathy between students and professors. For clinicians who work with patients, research on the development of empathy will be of great interest. The purpose of the research paper is well-stated, and the logical progression is well-written with no significant errors. In reviewing the paper, I found a few revisions.

  1. What does "total population" mean in the paragraph on lines 246-260? I'm not sure what it means. The term needs to be clarified.
  2. Figure 1 and Figure 2 are missing on lines 270 and 275.

I would also recommend adding a section in the Discussion section about teaching methods for empathy.

Thank you.